# The Psychosocial Impact of Insight Paradox and Internalized Stigma in Chronic Psychotic Disorders

**DOI:** 10.3390/bs15040410

**Published:** 2025-03-24

**Authors:** Juan Jesús Muños García, Ricardo M. Hodann-Caudevilla, Alfonso García Castaño, Sergio Aguilera Garrido, Rafael Durán Tischhauser, Álvaro Pico Rada, Rafael Salom

**Affiliations:** 1Centro de San Juan de Dios, Orden Hospitalaria San Juan de Dios, 28350 Ciempozuelos, Spain; juanjesus.munoz@sjd.es (J.J.M.G.); alvaro.pico@sjd.es (Á.P.R.); 2Fundación Hospitalarias Zaragoza, Hermanas Hospitalarias, 50190 Zaragoza, Spain; ricardo.hodann@fundacionhospitalarias.org; 3Hospital Infanta Sofía, 28702 San Sebastián de los Reyes, Spain; agcastano@salud.madrid.org; 4Centro de Salud Mental Parla, 28981 Parla, Spain; aguilera.g.sergio@gmail.com; 5Centro de Salud Mental Rivas-Vaciamadrid, 28529 Rivas-Vaciamadrid, Spain; rafael.duran@salud.madrid.org; 6Clínica Nuestra Señora de la Paz, Orden Hospitalaria San Juan de Dios, 28033 Madrid, Spain; 7Faculty of Psychology, Valencian International University (VIU), 46002 Valencia, Spain

**Keywords:** schizophrenia, insight paradox, internalized stigma, psychosocial outcomes, quality of life

## Abstract

Stigma and discrimination remain significant challenges to the quality of life and social integration of individuals with chronic mental disorders, particularly schizophrenia, one of the most stigmatized conditions. The paradox of insight, in which greater awareness of illness correlates with poorer psychosocial outcomes in the presence of high internalized stigma, provides a critical framework for understanding these challenges. This study examined the moderation between insight and internalized stigma and its influence on psychosocial outcomes in 83 male participants diagnosed with psychotic spectrum disorders. Using K-means clustering, three distinct profiles emerged: (1) good insight and minimal stigma, (2) poor insight and mild stigma, and (3) good insight and severe stigma. These profiles showed significant differences in depression, quality of life, and global functioning. Findings confirmed that internalized stigma moderates the relationship between insight and psychosocial well-being, exacerbating the negative influence of insight on quality of life and psychological health when stigma levels are high. The results emphasize the need for psychoeducational interventions to normalize experiences of psychosis, family and community engagement to reduce stigma, and cognitive-behavioral therapies to address stigma-related beliefs. These strategies are essential for improving psychosocial well-being and supporting recovery in this population.

## 1. Introduction

Concerning chronic mental disorders, stigma and discrimination remain significant challenges that directly affect individuals’ quality of life and social integration ([10]; [14]; [29]). This stigma, rooted in a combination of stereotypes, prejudices and discrimination, not only exacerbates the suffering of those experiencing these conditions but also hinders their access to employment opportunities, social relationships, and appropriate healthcare services ([16]). Schizophrenia, as one of the most stigmatized mental disorders, faces not only the inherent challenges of its condition but also the additional burden of social rejection and lack of understanding ([26]).

Within this context, the concept of insight has emerged as a crucial factor in understanding schizophrenia and guiding its clinical management. Defined as awareness of the illness, its symptoms, and the need for treatment, insight plays a complex role in shaping clinical and psychosocial outcomes ([16]; [26]). Beyond the awareness of one’s diagnosis and symptoms, insight also encompasses an understanding of the discrimination and stigma experienced in social interactions ([29]; [16]; [26]; [27]). Individuals with high insight are not only aware of their condition but also recognize the prejudices and differential treatment they face from others, which can significantly affect their self-esteem, hope, and quality of life. This dual dimension of insight—encompassing both illness awareness and social stigma—has garnered increasing interest from the scientific community over the past three decades due to its relevance in understanding the illness and its clinical management ([27]). Insight is defined as awareness regarding the symptoms associated with the disorder, its consequences, the need for treatment, and cognitive impairments ([18]). The limitations in achieving satisfactory quality of life despite advancements in antipsychotic treatments have been highlighted in the recent literature that systematically reviewed the effects of antipsychotic medications on quality of life in patients with schizophrenia, emphasizing that while second-generation and long-acting injectable antipsychotics show promising outcomes, quality of life often remains suboptimal due to persistent symptoms and unmet psychosocial needs, including stigma related ones ([24]). Insight is particularly relevant due to its significant role in understanding and managing schizophrenia, influencing both clinical and psychosocial outcomes ([9]).

Between 50% and 80% of patients with schizophrenia exhibit low levels of insight, primarily due to a disconnection from reality inherent to the disorder ([18]). Antipsychotic treatment helps to reduce psychotic symptoms and stabilize patients; however, it also increases insight, leading individuals to become more aware of their condition ([24]; [9]). While improved insight can enhance treatment adherence and psychosocial functioning, it may also have negative consequences, such as increased depression and lower quality of life ([4]; [7]; [21]; [15]; [23]). This phenomenon, known as the ‘insight paradox’, underscores the intricate balance between symptom management and psychological well-being in schizophrenia ([18]; [21]; [15]). Understanding the relationship between insight and quality of life is crucial to developing comprehensive interventions that combine pharmacological treatment with psychosocial and educational support, aiming to mitigate the potential adverse effects of increased insight while optimizing overall patient outcomes ([4]; [7]; [6]).

On the other hand, while insight refers to awareness of the illness and its implications, it can also lead to internalized stigma, which arises from how individuals interpret and respond to this insight ([2]; [8]). In 2007, Lysaker et al. proposed an explanation for the “insight paradox.” They suggested that the effects of insight vary depending on the meaning individuals with schizophrenia attribute to the disorder. Individuals perceiving schizophrenia as profoundly negative (those with high levels of internalized stigma) experience negative effects on psychosocial variables such as self-esteem, hope, and functioning when high insight is combined with internalized stigma. Conversely, individuals viewing the disorder as less negative—those with low internalized stigma—do not experience these negative effects with high insight. The authors concluded that individuals with mental disorders who possess good awareness of their condition yet internalize stigmatizing beliefs are at a heightened risk of low self-esteem and hopelessness, which could negatively impact their motivation toward achieving personal and/or rehabilitation goals.

Furthermore, other studies have yielded similar results, providing empirical support for the interaction between stigma and insight to explain the insight paradox ([15]; [3]; [5]; [11]; [13]). For example, Chio et al. conducted a study with patients diagnosed with psychotic spectrum disorders, measuring various variables over a year. Their results revealed that higher levels of insight were associated with lower life satisfaction, with stigma mediating this relationship: greater stigma intensified the negative effect of insight on life satisfaction.

This research aims to explore the complexities of insight and internalized stigma in individuals diagnosed with psychotic spectrum disorders. It examines the interaction between insight and stigma, highlighting the moderating role of stigma in the “insight paradox” and its mediating effect on quality of life, as well as other psychosocial variables such as depression and global functioning. In particular, this study focuses on individuals receiving long-term psychiatric care, a clinically distinct and underrepresented population in research on insight and stigma. This allows us to examine the paradox in a chronic and highly institutionalized context, where factors such as prolonged hospitalization and social isolation may uniquely moderate the relationship between insight and psychosocial outcomes.

## 2. Materials and Methods

To explore the moderation between insight and internalized stigma among individuals with psychotic spectrum disorders, the overarching aim of this study was to investigate these dynamics and assess their influence on various psychosocial outcomes. The specific objectives were: (1) to measure the moderation between internalized stigma and various aspects of quality of life and psychological health; (2) to identify distinct clusters within the sample based on levels of illness perception and internalized stigma; (3) to assess significant differences between clusters in terms of psychotic symptoms, stigma, quality of life, depression, vitality, adjustment, and optimization to gain a better understanding of the distribution of effects among patients; and (4) to conduct a moderation analysis to evaluate the influence of internalized stigma on the moderation between insight and quality of life, aiming to identify how these variables interact to influence psychosocial outcomes.

### 2.1. Participants

Eighty-three males diagnosed with psychotic spectrum disorders were included in the study. Diagnoses included paranoid schizophrenia (*n* = 56), residual schizophrenia (*n* = 23), and schizoaffective disorder (*n* = 4), which were confirmed using the Composite International Diagnostic Interview ([30]). Participants were recruited from the Prolonged Psychiatric Care Unit at the Centro San Juan de Dios in Ciempozuelos, Madrid, Spain. All participants were receiving their usual treatment under prolonged hospitalization and were in a stable phase of their psychopathological condition, defined by the absence of acute symptoms that could interfere with their understanding of participation in the study, as well as no changes in medication in the last month. Participants diagnosed with intellectual disability were excluded.

This study included a total of 83 male participants diagnosed with psychotic spectrum disorders, primarily paranoid or residual schizophrenia, who were in a stable clinical condition. The mean age of the participants was 47.22 years (SD = 9.75). They experienced their first hospitalization at approximately 21.12 years of age (SD = 4.81). On average, participants had undergone 10.85 hospitalizations throughout their clinical history (SD = 10.62). Additionally, the mean duration of stay in long-term care units was 8.25 years (SD = 9.01).

The sample consisted exclusively of male participants due to the low number of female inpatients in the long-term psychiatric care unit from which participants were recruited. Additionally, the inclusion criteria, which required clinical stability and the ability to complete the assessments, further restricted the potential participation of female patients.

Although the sample size may appear limited compared to larger-scale studies, it is important to consider that the target population comprises patients in prolonged psychiatric care units, a clinically specialized and less accessible subgroup. The participation rate, approximately 20% of eligible patients, reflects inherent challenges in recruiting this population, such as clinical instability or cognitive limitations.

### 2.2. Inclusion and Exclusion Criteria

Participants were selected based on the following criteria: inclusion criteria included a confirmed diagnosis of psychotic spectrum disorders according to the International Classification of Diseases, 10th Edition (ICD-10) ([31]), clinical stability defined as the absence of acute symptoms within the past month, and no changes in the prescribed medication regimen during the last month. Exclusion criteria included the presence of comorbid intellectual disability, acute symptoms that could impair comprehension or hinder participation in the study, and the inability to complete the required questionnaires independently or with minimal assistance. The low prevalence of women in this specific care unit, combined with the study’s exclusion criteria, significantly hindered their inclusion in the research.

### 2.3. Materials Design and Selection Procedure

Questionnaires were administered to assess various factors, including perception of discrimination, stigmatizing attitudes, global functioning, depression, and quality of life across multiple dimensions, with the aim of capturing a broad range of variables relevant to psychiatric research:Sociodemographic Data: Data related to various sociodemographic variables of the participants were collected. Information was requested regarding age, gender, marital status, level of education attained, current occupation, and any civil incapacitation and degree of disability.Positive and Negative Syndrome Scale (PANSS) ([22]): This scale, specifically designed for schizophrenia, assesses positive, negative, and insight symptoms through a semi-structured interview with 30 items. The insight item evaluates awareness of symptoms and the need for treatment. The PANSS includes subscales for positive symptoms (e.g., delusions, hallucinations), negative symptoms (e.g., affective flattening, social withdrawal), and general psychopathology (e.g., anxiety, depression). The total score ranges from 30 to 210, with each item scored from 1 (absent) to 7 (extreme). The PANSS Composite provides a balance measure by subtracting negative from positive symptom scores. Interrater reliability was high, with intraclass correlation coefficients ranging from 0.84 to 0.93.Internalized Stigma of Mental Illness Inventory (ISMI) ([1]): ISMI was applied to evaluate the subjective experience of stigma in patients. This questionnaire consists of 29 items exploring dimensions such as alienation, stereotype endorsement, perceived discrimination, and social isolation. The subscale of resistance to stigma was not used. Each item is rated on a 4-point Likert scale (1 = strongly disagree to 4 = strongly agree), with total scores ranging from 29 to 116. Higher scores indicate greater internalized stigma. The adapted version achieved strong internal consistency and test–retest reliability levels, with respective scores of 0.91 and 0.95 for the total scale.WHO Quality of Life-BREF (WHOQOL-BREF) ([17]): This instrument consists of 26 items grouped into four dimensions that assess the quality of life in different cultural contexts. Physical health evaluates an individual’s perceived energy levels, pain, sleep quality, and ability to perform daily activities. Psychological health assesses emotional well-being, self-esteem, cognitive function, and body image. Social relationships measure the quality and availability of social support, personal relationships, and satisfaction with social interactions. The environment reflects perceptions of safety, financial security, access to healthcare and leisure opportunities, and overall living conditions. Each item is rated on a 5-point Likert scale (1 = very poor/dissatisfied to 5 = very good/satisfied), with domain scores transformed to a 0–100 scale for easier interpretation. Internal consistency varied between domains, ranging from a Cronbach’s alpha of 0.69 (physical) to 0.90 (spirituality/religion/personal beliefs). Similarly, Cronbach’s alpha ranged from 0.74 (psychological) to 0.80 (physical).Calgary Depression Scale for Schizophrenia ([25]): This is a tool specifically designed to assess depression levels in individuals with schizophrenia, both during acute phases and deficit states, while distinguishing from positive, negative, and extrapyramidal symptoms. It consists of 9 items, each rated from 0 (absent) to 3 (severe), yielding a total score range of 0 to 27. It demonstrates high internal consistency, Cronbach’s alpha between 0.70 and 0.90.MOLDES ([12]): The MOLDES questionnaire consists of 87 items that assess the habitual way individuals face reality, cognitively and affectively. The questionnaire comprises three focal frameworks or third-order factors that encompass each person’s competencies in relating to reality: vital spontaneity, adjustment, and optimization, through which global functioning is evaluated. Vital spontaneity refers to an individual’s capacity for emotional expressiveness, enthusiasm, and openness to experiences. Adjustment reflects the ability to regulate emotions and behaviors in response to situational demands and social expectations. Optimization assesses the tendency to maximize personal growth, well-being, and adaptive resources over time. Participants indicate their degree of agreement or disagreement with each description on a 5-point Likert scale (1 = totally disagree to 5 = totally agree). The total score ranges from 87 to 435, with higher scores reflecting greater adaptation and resilience. The total reliability of the scale is Cronbach’s alpha = 0.90.

All procedures were approved by the Ethics and Research Committee of Hospital Universitario 12 de Octubre. At the onset of the recruitment period, clinical psychologists verified the eligibility of patients and provided them with a detailed explanation of the study. If patients agreed to participate, they were given an informed consent form to sign, and any queries they had were addressed. Additionally, participants were informed that their involvement was entirely voluntary and that they could withdraw consent at any time. Patients did not receive any financial incentives for participating in the study. Once informed consent was provided and agreement to participate confirmed, participants received the questionnaires to complete.

### 2.4. Statistical Analysis

For the analyses, Jamovi (Version 2.5) for Windows ([28]) was used. To achieve the objectives of this study, various analytical methodologies were employed, including internal consistency analysis, K-means clustering, ANOVAs, correlations, and moderation analysis.

Firstly, an internal consistency analysis of the ISMI scale was conducted to assess whether the first four subscales could be combined into a total score for cluster analysis and whether to include the fifth subscale, stigma resistance.

K-means clustering was used to identify distinctive patterns within the data by grouping participants based on their levels of insight and internalized stigma, allowing for a more detailed understanding of the heterogeneity within the sample. To ensure comparability and eliminate scale differences, scores were standardized into Z-scores. Additionally, ANOVAs were conducted to assess significant differences between the identified clusters across various psychosocial variables, such as psychotic symptoms, quality of life, and depression. Meanwhile, correlations examined linear relationships between internalized stigma, insight, and psychosocial outcomes.

Finally, moderation analyses were performed to explore how internalized stigma influences the relationship between insight and quality of life. These analyses focused on examining the interactions between internalized stigma and insight, enabling a more detailed assessment of their effects on quality of life. Bootstrap with 5000 samples was used to ensure robust estimates.

## 3. Results

Table 1 presents the descriptive statistics for key psychological and clinical variables assessed in the participants. These statistics provide an overview of central tendencies, variability, and range, offering a foundational understanding of the sample’s characteristics.

### 3.1. Internal Consistency Analysis

Firstly, we assume that internal consistency analysis of the internalized stigma scale (ISMI) item set was conducted. The Cronbach’s Alpha coefficient obtained was 0.904, indicating the high reliability of the instrument used. Additionally, Pearson correlations were performed among the five internalized stigma subscales: alienation, stereotype endorsement, perceived discrimination, social isolation, and stigma resistance. These correlation coefficients were used to measure the strength and direction of linear relationships between each pair of variables. The results reveal significant correlations among the first four subscales (*p* = 0.00), though not with the last subscale, stigma resistance, which was subsequently excluded from the total score.

### 3.2. Correlations Analysis

Additionally, correlations were conducted between insight and stigma scores with various psychological and quality of life measures, including depression level, vitality, adjustment, and optimization. Significant negative correlations were found between internalized stigma and several factors, such as psychological health (r = −0.251, *p* = 0.022), environmental (r = −0.288, *p* = 0.008), quality of life (r = −0.273, *p* = 0.013), adjustment (r = −0.255, *p* = 0.020), and vitality spontaneity (r = −0.218, *p* = 0.048). Furthermore, significant negative correlations were observed between depression level and psychological health (r = −0.294, *p* = 0.007) as well as quality of life (r = −0.254, *p* = 0.020), and social relationships (r = −0.295, *p* = 0.007). Moreover, optimization showed significant positive correlations with psychological health (r = 0.224, *p* = 0.042), environment (r = 0.341, *p* = 0.002), quality of life (r = 0.317, *p* = 0.004), and social relationships (r = 0.253, *p* = 0.021). Finally, significant positive correlations were found between vital spontaneity and physical health (r = 0.260, *p* = 0.018). The remaining correlations were not statistically significant.

### 3.3. Cluster and One-Way ANOVA Analysis

Continuing with the study objectives, the K-means clustering method was applied to perform cluster analysis on the dataset, focusing on insight and internalized stigma variables. This procedure identified three distinct groups among participants: Cluster 1, characterized by good insight and minimal stigma (*n* = 27), included participants who exhibited a high level of awareness regarding their condition combined with minimal internalized stigma. Cluster 2, defined by poor insight and mild stigma (*n* = 26), consisted of individuals with limited awareness of their condition and mild levels of internalized stigma. Finally, Cluster 3, marked by good insight and severe stigma (*n* = 30), included participants who displayed high levels of awareness of their condition but also significant internalized stigma.

ANOVA results (Table 2) showed significant differences between clusters for both, internalized stigma scores (F_(2, 80)_ = 56.57, *p* < 0.001) and positive, negative, and insight symptoms scores (F_(2, 80)_ = 62.36, *p* < 0.001).

Similar analyses were conducted for quality of life, depression, vital spontaneity, adjustment, and optimization scale results. ANOVA results indicated significant differences between clusters for all assessed variables, including physical health (F_(2, 80)_ = 20.37, *p* < 0.001), psychological health (F_(2, 80)_ = 40.16, *p* < 0.001), social relationships (F_(2, 80)_ = 36.73, *p* < 0.001), environmental (F_(2, 80)_ = 30.1, *p* < 0.001), quality of life (F_(2, 80)_ = 78.18, *p* < 0.001), depression (F_(2, 80)_ = 5.93, *p* = 0.004), vital spontaneity (F_(2, 80)_ = 16.25, *p* < 0.001), adjustment (F_(2, 80)_ = 10.76, *p* < 0.001), and optimization (F_(2, 80)_ = 14.63, *p* < 0.001) (Table 2).

### 3.4. Moderation Analysis

A moderation analysis was conducted to evaluate whether internalized stigma moderates the relationship between insight and quality of life. The model, which included insight, stigma, and the interaction term between these variables, revealed significant results for the moderation effect (see Figure 1).

The analysis showed that internalized stigma had a significant negative effect on quality of life (B = −0.240, SE = 0.115, Z = −2.084, *p* = 0.037), indicating that higher levels of internalized stigma are associated with a decrease in quality of life. However, insight did not exhibit a significant direct effect on quality of life (B = −0.086, SE = 0.172, Z = −0.498, *p* = 0.618).

Importantly, the interaction term between insight and stigma was significant (B = 0.032, SE = 0.013, Z = 2.578, *p* < 0.010), indicating that internalized stigma moderates the relationship between insight and quality of life. Specifically, the positive coefficient suggests that the negative impact of insight on quality of life decreases as levels of internalized stigma increase. The 95% confidence interval for the interaction term (CI95% = [0.00204–0.0531]) did not include zero, further supporting the significance of the moderation effect.

## 4. Discussion

The central aim of this study was to analyze the moderation between insight and internalized stigma in individuals with psychotic spectrum disorders, evaluating its influence on quality of life and other psychosocial outcomes, such as depression and emotional adaptation. Specifically, the study explored the moderating role of stigma in the paradox of introspection and quality of life. This research was conducted in a chronic and highly institutionalized population, a context in which factors such as prolonged hospitalization, reduced social contact, and loss of autonomy may significantly influence how patients process introspection and stigma.

First, the findings of this study indicate that high introspection does not necessarily predict better psychosocial outcomes. While patients with high introspection and low stigma (Cluster 1) exhibited higher levels of self-esteem, hope, and quality of life—suggesting that the absence of stigmatizing beliefs allows introspection to have a positive influence on psychosocial adaptation—this result aligns with previous research indicating that awareness of illness, when not accompanied by self-deprecation, can facilitate more adaptive coping strategies and encourage help-seeking behaviors ([19]).

On the other hand, in line with [21] ([21]), who reported an association between high introspection and elevated levels of depression, our results revealed that patients with high introspection combined with severe stigma (Cluster 3) experienced higher levels of depression. This suggests that, rather than facilitating adaptation, illness recognition may become emotionally disruptive when accompanied by a negative self-perception and an identity shaped by severe mental illness.

Similarly, individuals with high introspection and high stigma showed significant reductions in self-esteem, hope, and quality of life, indicating that illness awareness, when internalized within a stigmatizing framework, can contribute to emotional and functional deterioration. This finding is consistent with [2] ([2]), who suggested that, in individuals with schizophrenia, greater introspection may be associated with increased emotional vulnerability, as perceiving oneself as a person with a chronic mental illness may generate feelings of hopelessness and social withdrawal. Furthermore, they highlighted that introspection, without adequate support to reframe identity and life expectations, can reinforce negative rumination and a deteriorated self-perception, ultimately impacting functionality and social integration.

This pattern reinforces the insight paradox described by [18] ([18]), emphasizing that introspection can be a double-edged sword when combined with deeply internalized stigma. That is, while greater illness awareness may be beneficial in contexts where social support and adaptive coping strategies are available, in environments where internalized stigma is high, introspection may amplify psychological distress, reinforcing feelings of hopelessness, guilt, and social exclusion.

This phenomenon is particularly relevant in chronic institutionalized psychiatric populations, where social interaction is restricted, and prolonged exposure to stigmatizing narratives about mental illness may hinder the development of a resilient identity. In this context, the emotional and cognitive consequences of stigma emerge as key factors that can deepen the relationship between introspection and self-stigma. Previous research, such as that of [11] ([11]), has shown that psychological and emotional variables, such as proneness to self-criticism and negative self-evaluation, act as critical mediators in the relationship between introspection and self-stigma. These findings suggest that negative self-perception derived from internalized stigma may intensify in individuals with higher illness awareness, indicating that patients with high levels of self-criticism and self-deprecating thoughts tend to transform their awareness of illness into a sense of self-rejection. This mechanism may partly explain why, in our study, patients in Cluster 3—characterized by high introspection and severe stigma—exhibited a significant reduction in self-esteem and quality of life, along with increased depressive symptoms.

Furthermore, this study provides a novel contribution by examining the moderation of internalized stigma in the relationship between introspection and quality of life. Moderation analyses suggest that introspection does not exert a direct impact on quality of life; rather, its negative influence is amplified or mitigated depending on the degree of internalized stigma. In other words, the way in which a patient interprets their diagnosis and perceives their illness—either from a perspective of self-acceptance or self-rejection—determines whether introspection serves as an adaptive resource or an additional source of psychological distress.

Moreover, the results of this study indicate that the moderation between introspection and stigma influences quality of life, consistent with [3] ([3]), who identified stigma as a key moderator in the interaction between introspection and life satisfaction. Additionally, the findings suggest that internalized stigma not only influences the perception of illness but also affects how patients cope with its psychosocial consequences, emphasizing the need to address stigma in clinical interventions.

On the other hand, the results showed that introspection has a significant negative influence on quality of life, indicating that higher levels of introspection are associated with decreased quality of life. Although stigma alone did not show a direct significant influence, the moderation term between introspection and stigma was significant. This suggests that the negative influence of introspection on quality of life decreases slightly as levels of internalized stigma increase, possibly because the negative beliefs associated with stigma act as a buffer, mitigating the additional emotional burden that higher levels of introspection might otherwise impose. These findings complement the observations of [24] ([24]), who argued that despite advances in pharmacological treatments, the quality of life of patients with schizophrenia remains suboptimal due to factors such as stigma.

Additionally, it is crucial to recognize that internalized stigma is not a static variable, but rather a dynamic process shaped by past and present experiences. Following the perspective of [23] ([23]), some patients may have integrated introspection within a growth-oriented narrative, allowing them to accept their diagnosis without internalizing stigmatizing beliefs (Cluster 1). These individuals, by not perceiving themselves exclusively through the lens of their illness, seem to have developed coping strategies that help them maintain a cohesive identity and preserve a sense of self-efficacy and hope. In contrast, other patients may have become trapped in a narrative of loss and hopelessness, where introspection does not function as an integrative mechanism but rather as a constant reminder of their deterioration and limitations (Cluster 3). In these cases, awareness of illness does not translate into greater control over their well-being, but rather into an internalization of stigma that deepens feelings of worthlessness, guilt, and self-rejection.

From a clinical perspective, these findings have significant implications for the treatment of chronic psychiatric patients. Traditionally, many psychiatric interventions have emphasized enhancing introspection as a key therapeutic goal. However, this study suggests that introspection, in the absence of appropriate strategies to manage stigma, may not be beneficial.

Thus, interventions aimed at improving introspection must incorporate specific components to reduce internalized stigma and negative self-perception associated with mental illness. In particular, compassion-focused therapy (CFT) has been shown to be effective in reducing self-criticism and fostering a more positive self-image in individuals with severe mental illness ([5]). Likewise, mentalization- and metacognition-based interventions could help patients differentiate between adaptive introspection and self-deprecating rumination, facilitating healthier processing of identity and illness.

In terms of group interventions, psychoeducation should not only focus on providing information about illness but also actively reshape patients’ narratives about themselves and their diagnosis. Studies such as those by [20] ([20]) have demonstrated that group interactions that foster social validation play a key role in reducing internalized stigma.

However, these contributions must be considered considering the study’s limitations. The sample, drawn exclusively from long-term care units, limits the scope and generalizability of the findings to broader clinical populations or community-based settings. The cross-sectional nature of the study prevents the establishment of causal relationships between insight, stigma, and their psychosocial influence. Additionally, reliance on self-reported measures introduces potential biases, such as social desirability or recall inaccuracies, that could affect the robustness of the results. While the focus on specific psychosocial outcomes is valuable, it may have overlooked other critical domains, such as social connectedness, occupational functioning, or physical health.

Addressing these limitations presents opportunities for future research. Longitudinal studies are needed to unravel causal pathways and temporal dynamics between insight, stigma, and psychosocial variables. Expanding the study population to include diverse care settings and cultural contexts would enhance the applicability of the findings. Furthermore, investigating other moderating and mediating factors, such as coping strategies, family support, or neurocognitive functioning, could deepen our understanding of these complex relationships. Testing specific interventions, such as stigma reduction therapies and tailored psychoeducational programs, through rigorous randomized controlled trials would also be essential in translating theoretical knowledge into actionable clinical strategies.

Additionally, one important limitation of this study is the lack of female participants. The sample was composed exclusively of men due to the low number of women in the long-term psychiatric care unit. This limits the generalizability of our findings, as gender differences in insight, stigma, and psychosocial outcomes have been reported in previous research. Future studies should aim to include more diverse samples to better understand the potential moderating effects of gender in the insight paradox.

Ultimately, while this study lays a solid foundation for understanding the moderation between insight and stigma, it also highlights the need for continued research to refine theoretical models and develop evidence-based interventions. Reducing the burden of internalized stigma has the potential to significantly improve the quality of life, psychosocial resilience, and functional outcomes for individuals with psychotic spectrum disorders, paving the way for more effective and compassionate care.

## 5. Conclusions

Finally, this study provides a novel perspective by focusing on a homogeneous population of chronic patients in long-term care units. Although specific, this approach has allowed for the identification of distinct patterns that contribute to a better understanding of the moderation between insight, stigma, and emotional symptoms. Segmentation based on these factors facilitates the detection of specific needs and the development of more tailored interventions for each subgroup.

The analysis of the insight paradox reinforces the critical role of internalized stigma as a moderator in the relationship between insight and quality-of-life outcomes. Furthermore, the identification of clinically relevant subgroups highlights the importance of personalized approaches in the development of therapeutic strategies, addressing both insight and stigma to optimize patient well-being.

## Figures and Tables

**Figure 1 behavsci-15-00410-f001:**
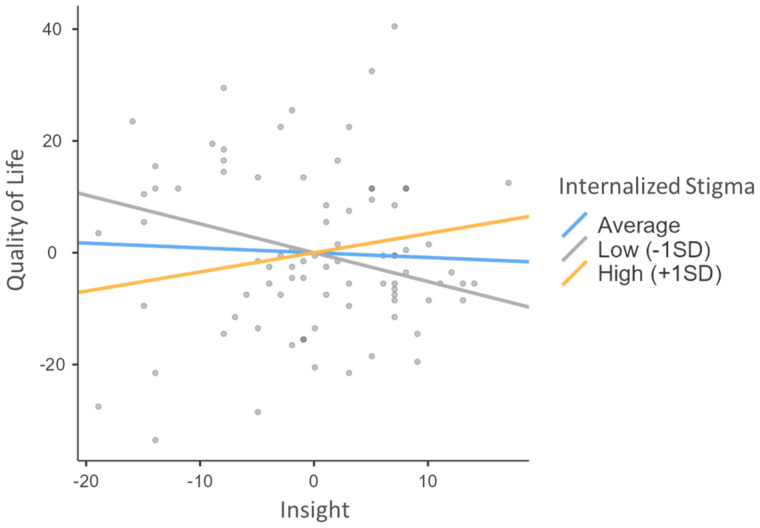
Moderation effects of insight and internalized stigma on quality of life.

**Table 1 behavsci-15-00410-t001:** Descriptive statistics.

	Mean	Standard Deviation	Min.	Max.
Stigma	65	13.4	32	97
Insight	3.46	0.80	2	5
Positive Symptoms	22.16	6.28	8	36
Positive–Negative Symptom Balance	−2.90	7.09	−19	10
Quality of life	89.51	14.32	56	130
Physical Health	24.40	3.89	14	35
Psychological Health	20.25	4.04	12	30
Social Relationships	9.46	2.94	3	15
Environment	28.77	5.36	15	40
Depression Level	4.16	5.10	0	21
Vital Spontaneity	58.21	11.20	34	86
Adjustment	61.31	10.39	27	96
Optimization	68.98	12.43	39	96

**Table 2 behavsci-15-00410-t002:** K-means cluster analysis and ANOVA of internalized stigma, insight, quality of life, depression level, vitality, adjustment, and optimization.

	Cluster 1	Cluster 2	Cluster 3	ANOVA	Comparison Group, *p* < 0.05
Internalized Stigma	−0.52	1.13	−0.51	56.57	2 > 1, 3 **
Insight	1.03	−0.11	−0.83	62.36	1 > 2, 3 **
Physical Health	0.97	−0.50	0.31	20.37	1 > 3 > 2 **
Psychological Health	1	−0.65	0.56	40.16	1 > 3 > 2 **
Social Relationships	0.81	−0.65	0.68	36.73	1 > 3 > 2 **
Environment	1.06	−0.57	0.39	30.1	1 > 3 > 2 **
Quality of Life	1.23	−0.73	0.57	78.18	1 > 3 > 2 **
Depression	−0.26	0.33	−0.44	5.93	2 > 1, 3 *
Vital spontaneity	−0.79	−0.14	0.75	16.25	3 > 2 > 1 **
Adjustment	−0.66	−0.12	0.64	10.76	3 > 2 > 1 **
Optimization	1.09	−0.19	−0.32	14.63	1 > 2, 3 **

Note: * *p* < 0.01; ** *p* < 0.001.

## Data Availability

The original contributions presented in this study are included in the article. Further inquiries can be directed to the corresponding author.

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
