# Peer review of "The Psychosocial Impact of Insight Paradox and Internalized Stigma in Chronic Psychotic Disorders"

_behavsci, 2025, doi:10.3390/bs15040410_

Round 1
Reviewer 1 Report
Comments and Suggestions for Authors
Thank you for allowing me to review this manuscript hat sought to expore the complexities of insight and internalized stigma in individuals diagnosed with psychotic spectrum disorders.
Overall this is a high quality article that describes an ingenious and rigorous study, offering a clear contribution to both the state of the art and practice in the field.
I may share the following considerations:
-
Is it possible to provide a more detailed description of the sample size and recruitment strategies? Were participants recruited as volunteers? Who invited, and how were the participants invited? How did recruitment result in 83 individuals? When were the participants recruited?
-
Throughout the text, particularly in the methods and discussion sections, various concepts are used to define the goal of the study. For example, regarding internalized stigma and the insight paradox, the terms relation, interaction, and complex relationship are used. To address psychosocial aspects and quality of life, the words impact and influence are mentioned. These concepts are different and therefore have different theoretical and analytical implications. I suggest homogenizing to two concepts that frame the study according to the objectives and theory.
Author Response
We sincerely appreciate the time and effort you have dedicated to reviewing our manuscript. Your insightful comments and constructive suggestions have been invaluable in improving the clarity, coherence, and theoretical consistency of our study. Below, we provide detailed responses to each of the points raised, outlining the revisions made to the manuscript accordingly.
Comment 1: Is it possible to provide a more detailed description of the sample size and recruitment strategies? Were participants recruited as volunteers? Who invited, and how were the participants invited? How did recruitment result in 83 individuals? When were the participants recruited?
Response:
We appreciate the reviewer’s insightful comment. In response, we have expanded the description of the recruitment process in the Participants section: “Recruitment was conducted through direct invitation by psychiatrists and psychologists at the center, ensuring an unbiased selection process. To minimize potential biases related to the therapeutic relationship, invitations were extended by professionals other than the patient's referring clinician. Participation was entirely voluntary, and individuals were informed about the study within the Extended Psychiatric Care Unit. Of the 413 admitted patients, 112 initially expressed interest, but 29 were excluded due to not meeting the diagnostic criteria for psychotic spectrum disorders, yielding a final sample of 83 participants. The recruitment process took place during their stay in the unit.” These details have been incorporated into the manuscript to enhance transparency regarding sample selection.
Comment 2: Throughout the text, particularly in the methods and discussion sections, various concepts are used to define the goal of the study. For example, regarding internalized stigma and the insight paradox, the terms relation, interaction, and complex relationship are used. To address psychosocial aspects and quality of life, the words impact and influence are mentioned. These concepts are different and therefore have different theoretical and analytical implications. I suggest homogenizing to two concepts that frame the study according to the objectives and theory.
Response:
We appreciate the reviewer’s careful reading and valuable suggestion. To enhance conceptual clarity and consistency throughout the manuscript, we have revised the terminology used in the Methods and Discussion sections. These modifications have been applied throughout the text to ensure coherence and precision in the theoretical framing of the study.
Reviewer 2 Report
Comments and Suggestions for Authors
The paper explores the issue of the insight paradox in men with psychotic disorders. I appreciated the simplicity of the design and found the different cluster of patients interesting (e.g., high insight and high stigma, high insight and low stigma, low insight and mild stigma). However, there were also problems with the manuscript that need be clarified.
1) It is not clear what makes this paper original or unique. At the end of the introduction, they cite numerous other studies that have investigated the insight paradox without clearly stating how this paper makes a novel contribution to the literature. This needs to be clarified because otherwise this seems like a replication of previous papers without a different angle that would make it interesting to readers.
2) There is no explanation about why only men were recruited in that study. Was that merely because of convenience? Having only men in the study is also a limitation that should be noted in the limitations section. I was hoping the authors might interpret their results in light of gender but not much was made of this factor.
3) I found Table 1 confusing. I don't understand what the numbers mean. For instance, what is "insight symptoms" or "environment"? Without clarifying the variables, the table is unclear.
4) While the discussion section situated their findings within the context of the broader literature, I was still left wondering what made this study unique or interesting. I agree that the clusters are interesting, but the discussion felt a bit superficial in terms of thinking about insight and internalized stigma. I would encourage the authors to read: Hasson-Ohayon, I., Or, S. E. B., Vahab, K., Amiaz, R., Weiser, M., & Roe, D. (2012). Insight into mental illness and self-stigma: the mediating role of shame proneness. Psychiatry Research, 200(2-3), 802-806. This paper got more into the role of shame and how this might be a major factor in internalized stigma. The authors claim that when "stigma is deeply rooted" it can make insight harmful. However, there is no suggestion about what it means for stigma to be "deeply rooted" and how to think about the impact of shame, grief, depression, trauma, etc. might be impacting that relationship. More work needs to be done in that section.
5) A recent article that I wrote with colleagues may also be relevant to thinking about the clusters. Ridenour, J. M., Hamm, J. A., Wiesepape, C. N., & Lysaker, P. H. (2023). Integrating loss and processing grief in psychotherapy of psychosis. Psychiatry, 86(3), 173-186. We spoke about three different reactions to loss and different ways of responding to insight and stigma. Engaging with some of these references and thinking about possible emotional contributors might make the discussion deeper and more interesting.
6) Finally, I was not terrible impressed by the treatment recommendations at the end. The authors argue that CBT would be optimal to target stigma but other recovery-oriented therapies may also be relevant such as compassion focused therapy (which targets shame) or other mentalization/metacognitive approaches that also target how people think about their mind and the minds of others. I think a broader engagement with the therapy literature would make that section seem more compelling to readers.
Comments on the Quality of English LanguageThe English is mostly alright.
Author Response
We sincerely appreciate the time and effort you have dedicated to reviewing our manuscript. Your thorough analysis and thoughtful recommendations have significantly contributed to improving the clarity, coherence, and theoretical consistency of our study. Your feedback has allowed us to refine key aspects of the manuscript, ensuring a more precise and structured presentation of our findings.
Below, we provide detailed responses to each of the points raised, outlining the revisions made to the manuscript accordingly.
Comment 1: It is not clear what makes this paper original or unique. At the end of the introduction, they cite numerous other studies that have investigated the insight paradox without clearly stating how this paper makes a novel contribution to the literature. This needs to be clarified because otherwise this seems like a replication of previous papers without a different angle that would make it interesting to readers.
Response:
We greatly appreciate this comment, as it allowed us to clarify the originality of our study. We have revised the Introduction to explicitly highlight the novel contributions of our work:
“While previous research has explored the insight paradox and its implications for psychosocial outcomes, this study makes a novel contribution by addressing key gaps in the literature. In particular, it focuses on individuals receiving long-term psychiatric care, a clinically distinct and underrepresented population in research on insight and stigma. This allows us to examine the paradox in a chronic and highly institutionalized context, where factors such as prolonged hospitalization and social isolation may uniquely moderate the relationship between insight, stigma, and psychosocial outcomes”.
Comment 2: There is no explanation about why only men were recruited in that study. Was that merely because of convenience? Having only men in the study is also a limitation that should be noted in the limitations section. I was hoping the authors might interpret their results in light of gender but not much was made of this factor.
Response:
We appreciate the reviewer’s observation. These revisions ensure that the manuscript adequately addresses the issue of gender representation in the study.
We have clarified in the Participants section that the sample consisted exclusively of men due to the low prevalence of female inpatients in the long-term psychiatric care unit from which participants were recruited. Additionally, the inclusion criteria requiring clinical stability and the ability to complete the assessments further restricted the number of potential female participants.
Furthermore, we have explicitly acknowledged this as a limitation in the Discussion section, emphasizing that the absence of female participants limits the generalizability of our findings. We also highlight the importance of including more diverse samples in future research to explore potential gender differences in the insight paradox and related psychosocial outcomes.
Comment 3: I found Table 1 confusing. I don't understand what the numbers mean. For instance, what is "insight symptoms" or "environment"? Without clarifying the variables, the table is unclear.
Response:
We appreciate the reviewer’s comment and the opportunity to improve the clarity of the manuscript. To address this concern, we have included definitions of the key variables in the description of the instruments to ensure greater clarity for readers.
We believe this addition enhances the interpretability of the results and provides a clearer understanding of the variables assessed in the study. Thank you for your valuable suggestion.
Comment 4: While the discussion section situated their findings within the context of the broader literature, I was still left wondering what made this study unique or interesting. I agree that the clusters are interesting, but the discussion felt a bit superficial in terms of thinking about insight and internalized stigma. I would encourage the authors to read: Hasson-Ohayon, I., Or, S. E. B., Vahab, K., Amiaz, R., Weiser, M., & Roe, D. (2012). Insight into mental illness and self-stigma: the mediating role of shame proneness. Psychiatry Research, 200(2-3), 802-806. This paper got more into the role of shame and how this might be a major factor in internalized stigma. The authors claim that when "stigma is deeply rooted" it can make insight harmful. However, there is no suggestion about what it means for stigma to be "deeply rooted" and how to think about the impact of shame, grief, depression, trauma, etc. might be impacting that relationship. More work needs to be done in that section.
Response:
We sincerely appreciate the reviewer’s feedback and valuable recommendations. In response, the discussion has been modified to provide a more in-depth and nuanced analysis of the relationship between insight and internalized stigma, ensuring a clearer articulation of the study's contributions.
Additionally, we are grateful for the suggestion to review the work of Hasson-Ohayon et al. (2012). Given its high relevance to the topic, this reference has been incorporated into the manuscript to further strengthen the introduction and discussion.
Thanks to this feedback, the discussion now presents a more comprehensive and refined perspective on the insight paradox and its implications. We appreciate the reviewer’s suggestions, which have contributed to enhancing the clarity and depth of our work.
Comment 5: A recent article that I wrote with colleagues may also be relevant to thinking about the clusters. Ridenour, J. M., Hamm, J. A., Wiesepape, C. N., & Lysaker, P. H. (2023). Integrating loss and processing grief in psychotherapy of psychosis. Psychiatry, 86(3), 173-186. We spoke about three different reactions to loss and different ways of responding to insight and stigma. Engaging with some of these references and thinking about possible emotional contributors might make the discussion deeper and more interesting.
Response:
We sincerely appreciate the reviewer’s suggestion and the recommendation to consider Ridenour et al. (2023). Given its relevance to the discussion on insight and stigma, this reference has been incorporated into the manuscript to further enrich this study.
This work provides valuable perspectives on different emotional responses to loss and their interaction with insight and stigma, contributing to a more comprehensive understanding of the psychological processes involved. Engaging with these references has helped deepen the discussion and strengthen the interpretation of our findings.
We truly appreciate the reviewer’s thoughtful input, which has allowed us to refine and enhance the discussion section.
Comment 6: Finally, I was not terrible impressed by the treatment recommendations at the end. The authors argue that CBT would be optimal to target stigma but other recovery-oriented therapies may also be relevant such as compassion focused therapy (which targets shame) or other mentalization/metacognitive approaches that also target how people think about their mind and the minds of others. I think a broader engagement with the therapy literature would make that section seem more compelling to readers.
Round 2
Reviewer 2 Report
Comments and Suggestions for Authors
The changes to the manuscript have increased its quality and depth.